# Regulation of PGC-1α of the Mitochondrial Energy Metabolism Pathway in the Gills of Indian Medaka (*Oryzias dancena*) under Hypothermal Stress

**DOI:** 10.3390/ijms242216187

**Published:** 2023-11-10

**Authors:** Naveen Ranasinghe, Wei-Zhu Chen, Yau-Chung Hu, Lahiru Gamage, Tsung-Han Lee, Chuan-Wen Ho

**Affiliations:** 1Department of Life Sciences, National Chung Hsing University, Taichung 402, Taiwan; madhumalranasinghe@gmail.com (N.R.);; 2The iEGG and Animal Biotechnology Center, National Chung Hsing University, Taichung 402, Taiwan; 3International Master’s Program of Biomedical Sciences, College of Medicine, China Medical University, Taichung 402, Taiwan

**Keywords:** Indian medaka, salinity, cold-stress, cold-tolerance, mitochondria source, gill, PGC-1α

## Abstract

Ectothermic fish exposure to hypothermal stress requires adjusting their metabolic molecular machinery, which was investigated using Indian medaka (*Oryzias dancena*; 10 weeks old, 2.5 ± 0.5 cm) cultured in fresh water (FW) and seawater (SW; 35‰) at room temperature (28 ± 1 °C). The fish were fed twice a day, once in the morning and once in the evening, and the photoperiod was 12 h:12 h light: dark. In this study, we applied two hypothermal treatments to reveal the mechanisms of energy metabolism via *pgc-1α* regulation in the gills of Indian medaka; cold-stress (18 °C) and cold-tolerance (extreme cold; 15 °C). The branchial ATP content was significantly higher in the cold-stress group, but not in the cold-tolerance group. In FW- and SW-acclimated medaka, the expression of genes related to mitochondrial energy metabolism, including *pgc-1α*, *prc*, *Nrf2*, *tfam*, and *nd5,* was analyzed to illustrate differential responses of mitochondrial energy metabolism to cold-stress and cold-tolerance environments. When exposed to cold-stress, the relative mRNA expression of *pgc-1α*, *prc*, and *Nrf2* increased from 2 h, whereas that of *tfam* and *nd5* increased significantly from 168 h. When exposed to a cold-tolerant environment, *prc* was significantly upregulated at 2 h post-cooling in the FW and SW groups, and *pgc-1α* was significantly upregulated at 2 and 12 h post-cooling in the FW group, while *tfam* and *nd5* were downregulated in both FW and SW fish. Hierarchical clustering revealed gene interactions in the cold-stress group, which promoted diverse mitochondrial energy adaptations, causing an increase in ATP production. However, the cold-tolerant group demonstrated limitations in enhancing ATP levels through mitochondrial regulation via the PGC-1α energy metabolism pathway. These findings suggest that ectothermic fish may develop varying degrees of thermal tolerance over time in response to climate change. This study provides insights into the complex ways in which fish adjust their metabolism when exposed to cold stress, contributing to our knowledge of how they adapt.

## 1. Introduction

There is a close relationship between the environment and teleostean fish. As ectotherms, fish adjust their body temperature in response to environmental temperatures [1,2]. Water temperature influences various biological processes in fish, including appetite, digestion, growth, immunological function, progression of specific diseases in fish, involves pathogen transfer, which influences disease development, ionic and osmotic regulation and energy metabolism [3,4,5,6,7,8]. Daily and seasonal temperatures have a significant impact on ectotherms, leading to the greatest influence on energy budgets, spawn times and their biogeographical dispersion and resulting in changes in evolutionary time [9,10]. Hypothermal stress-induced differential expression profiles of immune response genes in cold-acclimated euryhaline milkfish (*Chanos chanos*) after exposure to low temperatures at 18 °C [11,12], European sea bass (*Dicentrarchus labrax*) at 24–18 °C over one week [12] and seawater (SW) yellow drum (*Nibea albiflora*) subjected to cold stress at 8 °C lasted 30 days [13].

Being exposed directly to external environments, the gill is one of the primary organs for osmoregulation in fish and is sensitive to physical and chemical changes in aquatic environments such as temperature, acidification of water supplies, salts and heavy metals and water borne toxicants [14,15]. The gill has several functions, including gas exchange and osmoregulation [16,17]. It has been claimed that fish osmoregulation requires a substantial amount of metabolic energy [18,19]. Variation of water temperature increasing the amoebic gill disease causes gill pathology [20]. During cold water normoxic conditions, gill plasticity can lead to changes in ion flux and oxygen uptake by altering the surface area of gill lamellae, which are primarily embedded in an interlamellar cell mass [21,22]. In acclimation to environments of different salinities, the plasma glucose and lactate levels of flounder (*Solea senegalensis*) were reported to change significantly over time [23]. When exposed to higher-salinity environments, the activity of Na^+^/K^+^-ATPase (NKA), the primary driving force for osmoregulation, in the gills of the gilthead seabream (*Sparus aurata*) is altered with the metabolic accumulation of glycogen, glucose and lactate [24]. Energy supply is required in fish gills to maintain osmotic homeostasis through the energy consumptive active ion-transport system [25]. To support energy-dependent transepithelial ion transport in the gills, euryhaline Mozambique tilapia (*Oreochromis mossambicus*) was found to use glycogen phosphorylase, which catalyzes glycogen degradation in gill epithelial cells and aids in the rapid mobilization of local energy stores [26]. Tilapia gills are also capable of using the creatine kinase conversion mechanism (ADP + phosphocreatine → ATP + creatine) to provide energy for short-term requirements [27] for driving the NKA activity and the following ion transport system [27]. Moreover, ultrastructural investigations have revealed the existence of mitochondria-rich cells (MRCs) as a fundamental component within the surface layer of the gill epithelium [28]. The glucose transporter was found to be expressed in the gills and located in mitochondria-rich ionocytes, allowing for energy translocation in the gills [29]. Mitochondria are the primary sites for ATP synthesis, apoptosis initiation and regulation via important ions (e.g., calcium and potassium) for their metabolism [30]. The stimulation of mitochondrial biogenesis and thermogenesis is involved in the control of energy metabolism [31]. In response to cellular metabolic conditions, stress and other intracellular or external cues, mitochondrial biogenesis is regulated to support energy metabolism [32]. More studies are required to address the dearth of information and the inherent ambiguity in understanding how mitochondrial biogenesis and energy metabolism intersect in euryhaline fish during cold exposure, emphasizing the pressing need to investigate the long-term effects of hypothermal stress on fish gill mitochondria.

With the coordination of gene expression in energy metabolism regulation, several nuclear and mitochondrial encoded genes cooperate to respond to the different physiological stressors changing mitochondrial content, which is highly regulated by essential DNA binding proteins such transcriptional co-activators interact with transcriptional factors as nuclear respiratory factors (NRFs), which needs to control the functional regulation in mitochondria through mitochondrial transcriptional factors such as TFAM [33]. Peroxisome proliferator-activated receptor coactivator-1 (PGC-1) is a regulator of the mitochondrial biogenesis pathway in both fish and mammals [33,34]. It controls cellular energy transduction/energy production and has been widely described as a master regulator of mitochondrial biogenesis in mice [35,36] and zebrafish larvae (*Danio rerio*; [37,38]). PGC-1 has the ability to strongly stimulate mitochondrial activity in various organs, such as the heart and skeletal muscles, under specific circumstances, resulting in high oxidative capacity in mammals [39]. PGC-1, which encodes two isoforms, PGC-1α and PGC-1β, is involved in the synthesis and energy metabolic activity of the mitochondria in higher organisms [40]. When goldfish (*Carassius auratus*) were subjected to temperature changes (4–35 °C) with different diets (low fat and high fat), transcriptional levels for metabolic enzymes and mitochondrial enzyme activity were significantly correlated with *pgc-1α* and *pgc-1β* expression [33]. PGC-1α serves as a central regulator of mitochondrial biogenesis and energy production by partnering with deacetylases and phosphorylases to bind nuclear receptors [41]. This collaboration results in the upregulation of various transcription factors and subsequent expression of proteins encoded by both nuclear and mitochondrial genomes, particularly those associated with the oxidative phosphorylation complexes (OXPHOS) in the mitochondria [42]. This orchestration, led by PGC-1α, promotes the efficient production of ATP, a crucial factor in energy generation, and underlines its role as a master regulator of this process. Furthermore, PGC-1α’s activity is closely intertwined with AMPK, a sensor of cellular energy status, which further emphasizes its critical role in regulating mitochondrial biogenesis and oxidative metabolism [41]. Song, et al. [41] reported that siRNA knockdown of hepatic *pgc-1α* expression did not affect the mitochondrial DNA (mtDNA) copy number in blunt snout bream. However, it is still unclear how low temperatures influence *pgc-1α* expression in euryhaline fishes.

Furthermore, PGC-1 has been linked to coactivators, such as the PGC-1-related coactivator (PRC), as well as some other important transcriptional factors [40,43]. PGC-1β and PRC were reported to have overlapping roles with functional specialization in triggering transcription machinery in sunfish (*Centrarchidae*), pumpkinseed (*Lepomis gibbosus*), bluegill (*Lepomis macrochirus*), and black crappie (*Pomoxis nigromaculatus*; Bremer and Moyes [44]. PRC was found to resemble PGC-1α in structure and had an NH_2_ terminal region to aid in the dynamic transcriptional activation of mitochondrial biogenesis and respiratory function [45,46]. PRC binds to nuclear transcription factors including the nuclear respiratory factor (Nrf) and is involved in mitochondrial activity, functioning similarly to PGC-1 [46]. PGC-1α and PRC interact differently with nuclear hormone receptors, but they both directly bind to several nuclear transcription factors involved in respiratory chain expression [47]. However, knowledge of the regulatory mechanisms of fish PRC remains limited.

Mammals and fish have been found to contain Nrfs, including Nrf1 and Nrf2, which are crucial transcription factors involved in the mitochondrial acid cycle, induced by changes in water temperature [48]. Nrf2 has been reported to regulate oxidative stress by targeting antioxidant enzymes and cellular genes for defense against damage to the mitochondrial structure and functions in grass carp (*Ctenopharyngodon idella*; [49]). The Nrf2-TFAM pathway is activated in response to oxidative stress, which improves mitochondrial dysfunction and promotes mitochondrial biosynthesis in humans [50]. In non-stressed cells, Nrf2 is suppressed by Keap1, promoting its degradation, while in stressed cells, the disruption of this interaction activates Nrf2, enabling it to enter the nucleus and function as a transcriptional activator by binding to antioxidant response elements (AREs) in the DNA, thus allowing the transcriptional activation of its target genes [51,52]. Moreover, Nrf2 plays a role in the activation of mitochondrial biogenesis by binding to adenosine and uridine-rich elements (AREs) in the NRF1 promoter, ultimately leading to the activation of mitochondrial transcription factor A (TFAM) [53,54]. However, for energy homeostasis, the nuclear mitochondrial biogenesis program augmented the copy number of mitochondrial DNA in genome maintenance and mitochondrial biogenesis in mice by activating the nuclear regulation of *Nrf2* with *prc* to target genes via the transcription factor *tfam*, which is required for mtDNA replication [46,55,56]. When largemouth bass (*Micropterus salmoides*) were exposed to hypoxia and Cu^2+^ contamination, endoplasmic reticulum (ER) stress and mitochondrial damage induced gill apoptosis via the caspase activity pathway, while *tfam* expression was downregulated [57]. Splice-modifying morpholinos resulted in the knockdown of zebrafish embryonic *tfam*, a decrease in the mtDNA copy number and a decrease in ATP production [58]. Moreover, the absence of TFAM in mouse kidney epithelial cells leads to dramatically reduced mitochondrial gene expression, mitochondrial depletion and nephron maturation failure [59].

Zebrafish (*Danio rerio*); contained are 13 protein genes, 22 tRNAs 2 rRNAs and a noncoding control region was reported [60]. As a gene of mitochondrial respiratory complexes, ND5 was found to be downregulated in cold-stored embryos, reducing hatchability due to increased apoptosis in blue-breasted quails (*Coturnix coturnix*; [61]). There is still a huge discrepancy in mitochondrial TFAM regulation via ND5 in fish gills, thereby hindering the comprehensive understanding of the TFAM mechanisms in cold environments.

The euryhaline Indian medaka (*Oryzias dancena*) is a model species, known for its adaptability to changing environmental conditions, has been extensively studied for its responses to environmental stressors, encompassing factors such as salinity and temperature [14,62,63,64]. Thermal acclimation between 10 and 35 °C indicated strong thermal tolerance across both low and high temperature ranges [65].Notably, in previous research, marine Indian medaka larvae exhibited elevated mortality rates at low temperatures, along with indications of thermal discomfort, increased anaerobic metabolism and upregulated heat shock proteins in response to the cold, underscoring their reliance on ATPase activity for cellular protection [62,66]. However, the thermal adaptability of Indian medaka showed a decrease in temperatures, implying that the fish could enhance its thermal tolerance through thermal acclimation [62] and also in several fish species, as tropical yellowtail catfish (*Pangasius pangasius*) [67] and Climbing perch (*Anabas testudineus*) [68]. Accordingly, this study used this model species to investigate the changes in gill energy metabolism under hypothermal stress at various levels by target genes in the regulatory pathways, including *pgc-1α*, *prc*, *Nrf2*, *tfam*, and *nd5*. Hierarchical clustering was used to further illustrate the integration of gene expression profiles with comparable patterns over time to determine their role in ATP production. To uncover potential hypothermal stress regulatory mechanisms, our research explores the intricate post-transcriptional regulation governing energy metabolism in euryhaline teleosts, focusing on pgc-1α-related key mechanisms for both cold-stress and cold-tolerance in fish. With the significant economic impact of low-temperature tolerance in aquaculture and the growing interest in understanding post-transcriptional levels for predictive tools, our study provides support and guidance for future research, offering insights into potential targets for strategies involving aquatic organisms.

## 2. Results

### 2.1. ATP Content

The ATP content in the gills of (fresh water) FW- and (seawater) SW-acclimated Indian medaka was measured after 7 days (168 h) of exposure to 28, 18, and 15 °C. The cold-stress (18 °C) group had significantly higher ATP content in both FW and SW medaka than in the control (28 °C) group. However, the average ATP content of gills from the cold-tolerant (15 °C) group was not significantly higher than that of the control group (Figure 1A,B).

### 2.2. Differential mRNA Expression of Related Genes between FW and SW Medaka Gills under Cold-Stress (18 °C) Environments

Time-course effects of cold-stress (18 °C) on the mRNA expression of two nuclear transcription coactivators and two transcription factors were assayed in gills of FW- and SW-acclimated Indian medaka. The relative mRNA abundance of peroxisome proliferator-activated receptor coactivator (PGC)-1α increased significantly in both FW and SW medaka gills at 12 h and 2 h post-cooling, respectively (Figure 2A,B). However, 48 h post-cooling at 18 °C, the expression in FW and SW individuals steadily increased and decreased, respectively, resulting in significantly higher expression at 168 h in FW and aligned to the control group in SW groups (Figure 2A,B). The mRNA expression of the other coactivator, PGC-1-related coactivator (*prc*), was significantly higher with an acute response from 2 h after exposure to 18 °C from 28 °C in both the FW (Figure 2C) and SW groups (Figure 2D). Under cold-stress, significantly higher expression levels were maintained for 168 h (Figure 2C,D).

As a transcription factor, the relative mRNA abundance of nuclear respiratory factor 2 (*Nrf2*) surged at 24 h post cooling in FW (Figure 2E). In the SW cold-stress group, *Nrf2* expression was significantly upregulated at 12 h and 168 h post-cooling (Figure 2F). By interacting with the mitochondrial genome of the gills, the relative mRNA levels of mitochondrial transcription factor A (*tfam*) reached significantly higher levels within 168 h after exposure to 18 °C in both FW- and SW-acclimated medaka (Figure 2G,H). In the SW group, *Nrf2* and *tfam* responded early in the first 2 h to achieve significantly lower expression (Figure 2F,H). However, in the mitochondrial genome, the mitochondrial encoded respiratory protein, NADH dehydrogenase 5 (nd5), revealed similar patterns in mRNA expression as *tfam* increased significantly only in 168 h in the gills of both FW and SW individuals (Figure 3A,B).

### 2.3. The Hierarchical Cluster Analysis of the Cold-Stress Groups

The relationships among the expression of the five genes in the cold-stress groups were clearly visualized by the heat map. The individual tiles or rectangles in a heat map are scaled with a range of colors proportionate to gene expression values on a red-green scale (from lower to higher mRNA expression). A summary of the qPCR data and the interaction of gene expression in the cold-stress groups also showed different gene clusters with similar expression characteristics as [nd5, tfam; with lower divergence] and [(pgc1α, prc) Nrf2 with higher divergence] in FW-acclimated medaka (Figure 4A), and [Nrf2, pgc1α] and [(nd5, prc) tfam with higher divergence] in SW individuals (Figure 4B). 

### 2.4. Differential mRNA Expression of Related Genes between FW and SW Medaka under Cold-Tolerance (15 °C) Environments

When the Indian medaka were exposed to the cold-tolerant environment (15 °C), the mRNA expression of *pgc-1α* (Figure 5A,B), *prc* (Figure 5C,D), and *Nrf2* (Figure 5E,F) of the FW and SW groups did not significantly increase, except at 2 h and 12 h post-cooling of *pgc-1α* in FW (Figure 5A) and 2 h post-cooling of *prc* in FW and SW medaka (Figure 5C,D). In both the FW and SW groups, the relative mRNA expression of *tfam* decreased 2 h after post-cooling and returned to its initial level 48 h post-cooling (Figure 5G,H), although significance was only found in the FW medaka (Figure 5G). In addition, in the FW group, the mRNA expression in the mitochondrial genome of *nd5* decreased until it reached a significant level at 168 h (Figure 6A), whereas in the SW group, it increased significantly at 48 h and returned to the initial levels at 168 h post-cooling (Figure 6B).

**Figure 4 ijms-24-16187-f004:**
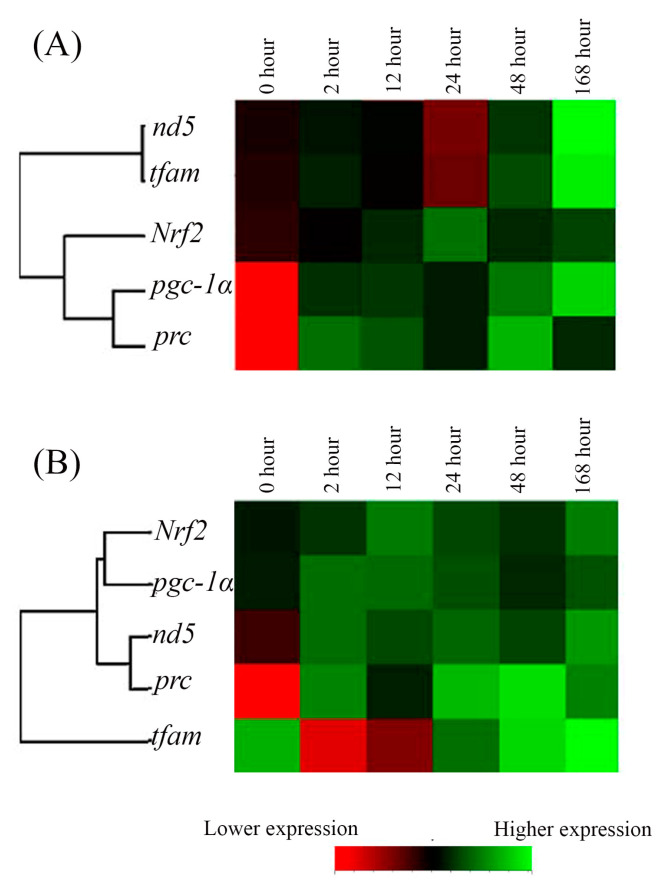
The time-point expression pattern cluster of mitochondrial energy metabolism-related genes in gills of the (**A**) fresh water (FW)- and (**B**) seawater (SW)-acclimated Indian medaka after exposure to 18 °C from 28 °C. The analyzed genes were hierarchically clustered. Red to green colors indicated low to higher gene expression.

### 2.5. The Hierarchical Cluster Analysis of the Cold-Tolerance Groups

The relationships among the expression of these five genes in Indian medaka at 15 °C in the FW- and SW-acclimated groups were compared. In FW- and SW-acclimated medaka, gene clusters were clearly visualized in hierarchical clusters: [*nd5, tfam* with higher divergence], [(*Nrf2, pgc1α*) *prc*] in the FW group (Figure 7A), and [*Nrf2* with higher divergence to (*nd5, tfam* with lower divergence) (*pgc1α, prc* with lower divergence)] in the SW group (Figure 7B).

**Figure 5 ijms-24-16187-f005:**
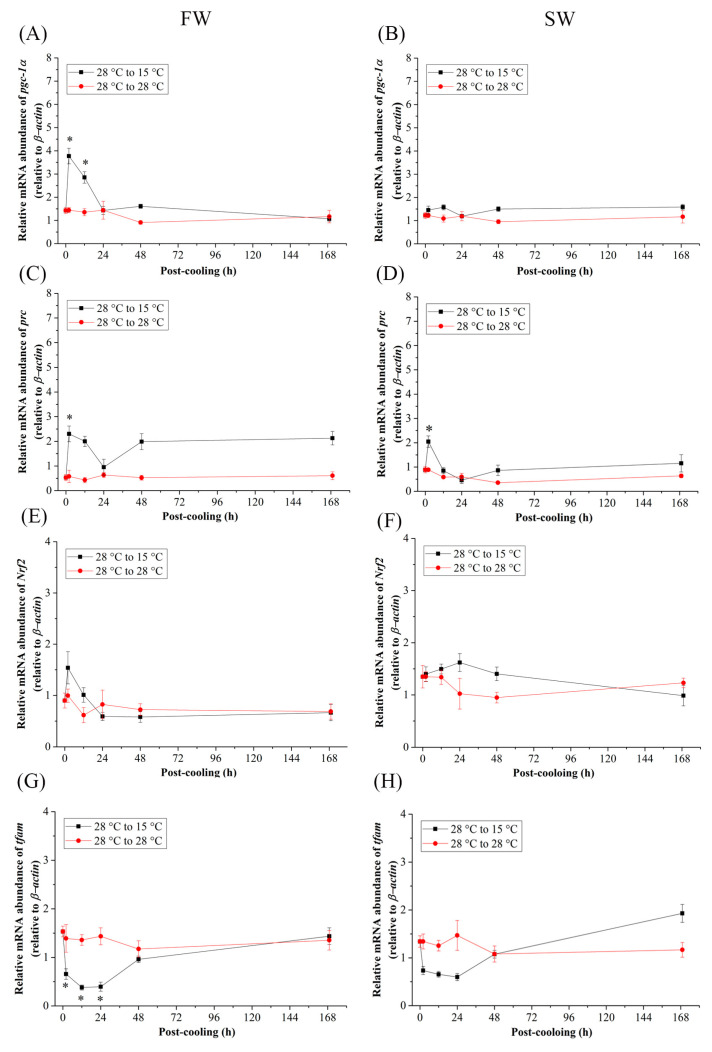
Relative mRNA expression of (**A**,**B**) *pgc-1α*, (**C**,**D**) *prc*, (**E**,**F**) *Nrf2*, and (**G**,**H**) *tfam* in gills of fresh water-(FW) and seawater (SW)- acclimated Indian medaka transferred from 28 °C to 28 °C, and cooled down at 15 °C for one week. The expression has been normalized by the expression of *β-actin*. The asterisks (*) indicate a significant difference at each time point in the control group. (*n* = 6; Mann-Whitney test, *p* < 0.05).

**Figure 6 ijms-24-16187-f006:**
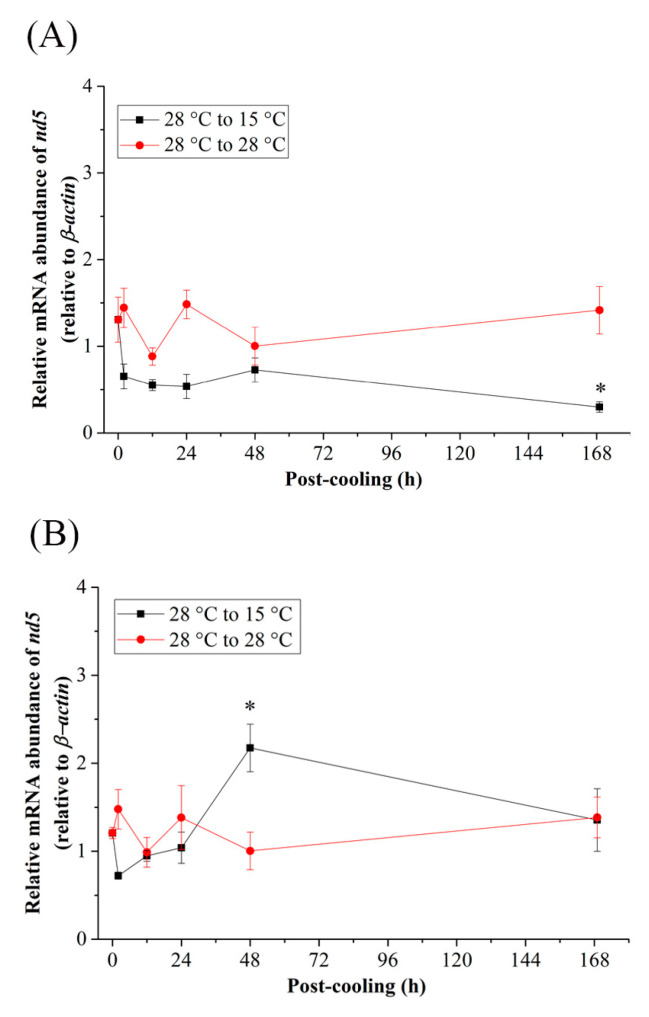
Relative mRNA expression of *nd5* in the gills of (**A**) fresh water (FW)- and (**B**) seawater (SW)-acclimated Indian medaka transferred from 28 °C to 28 °C, and cooled down at 15 °C for one week. The expression has been normalized by the expression of the *β-actin*. The asterisks (*) indicate a significant difference at each time point in the control group. (*n* = 6; Mann-Whitney test, *p* < 0.05).

**Figure 7 ijms-24-16187-f007:**
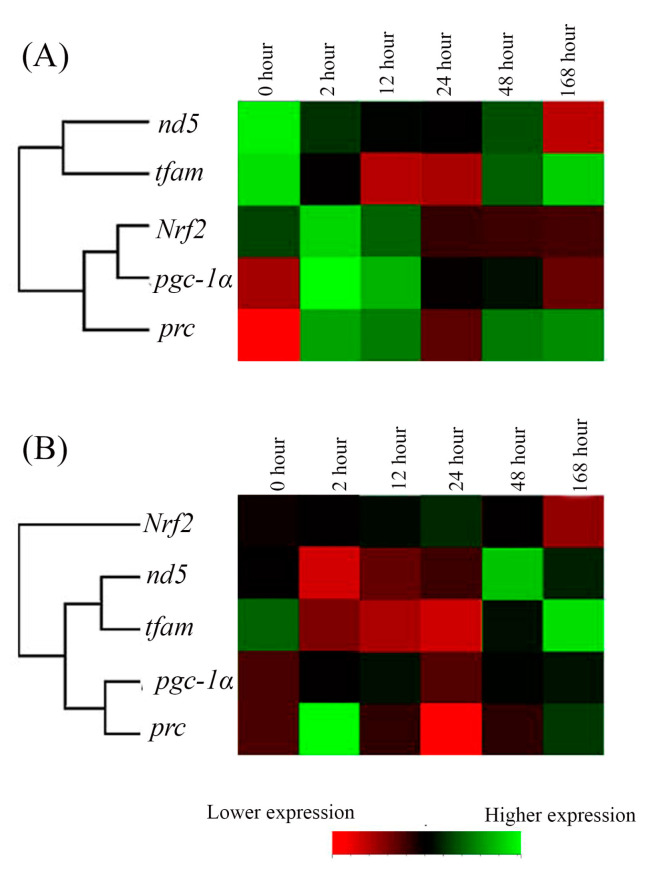
The time-point expression pattern cluster of mitochondrial energy meatabolism-related genes in the gills of the (**A**) fresh water (FW)- and (**B**) seawater (SW)-acclimated Indian medaka after exposure to 15 °C from 28 °C. The analyzed genes were hierarchically clustered. Red to green colors indicated low to higher gene expression.

## 3. Discussion

In this study, gills of both FW- and SW-acclimated Indian medaka exhibited a significant increase in ATP levels 168 h after transfer to the 18 °C environment, indicating an increasing energy demand for physiological roles in the gills under hypothermal stress. Climate change is expected to challenge ectotherms’ ability to function optimally [69], biochemical mechanisms have been reported to be involved in enhancing the adaptive ability of fish against cold stress in different cold temperatures, for example, the large yellow croaker (*Larimichthys crocea*; 15–7 °C [70]) and zebrafish larvae (16–12 °C; [71]). In SW-acclimated Indian medaka reared at different temperatures, the lowest mortality rate was observed in fish reared at 20–15 °C [62], based on Q10 values of physiological rates/variables were quantified in several teleosts at different temperatures [72,73,74]. Increased mitochondrial cholesterol content in Chinese hamsters has been shown to disrupt mitochondrial function and energy metabolism via reduced glutamine-dependent mitochondrial respiration and reduced ATP transport through mitochondrial membranes [75]. Our previous study revealed that there was a significant effect on biochemical mechanisms in the cholesterol accumulation in livers that were downregulated in SW- and upregulated in FW-acclimated Indian medaka when exposed to 18 °C for 168 h [64]. Wen, et al. [64] reported that the blue discus (*Symphysodon aequifasciatus*) could survive well at ambient temperatures ranging from 25.1 °C to 32.8 °C. When acclimated to a lower temperature of 20 °C, they started to reserve energy (lipids and carbohydrates) in the muscle. Meanwhile, their energy consumption increased, while cellular energy allocation, oxygen consumption rate, and energy metabolism-related enzyme activities decreased [76]. In addition, the gills of the blue discuss showed elevated levels of reactive oxygen species (ROS) and antioxidant defense responses, as well as modified metabolic pathways at low temperatures (20 °C; [77]). In transcriptome analyses of gills of genetically improved farmed tilapia (GIFT; *Oreochromis niloticus*) under cold-stress, genes related to the citric acid cycle were upregulated [78]. Within 48 h of cold (15 °C) exposure, the FW teleost *Astyanax lacustris* showed increased glycolysis, citric acid cycle, and amino acid catabolism. Meanwhile, oxidative stress is increased by lipid peroxidation and liver oxidative damage [79]. Although Indian medaka, like other teleosts, are susceptible to hypothermal stress, this study revealed that individuals appeared to survive by energy conversion in the cold-stress group at 18 °C.

Fish metabolism and energy requirements are influenced by factors like salinity and temperature. Most fish thrive in lower salinity ranges, which often result in lower metabolic rates [80,81]. For instance, rainbow trout in FW show better energy budgets due to optimal osmoregulation, while FW fish can struggle with energy allocation when salinity changes [81]. In another study, cold stress at 18 °C affected milkfish differently depending on their salinity adaptation. Freshwater milkfish experienced glycogen breakdown in their livers, an essential step in maintaining energy balance during cold stress, while both FW and SW milkfish increased anaerobic glycolysis in their muscles [82,83]. The study also found that cold stress increased energy demand and ATP production in medaka gills, regardless of the salinity condition. These findings demonstrate the complex relationship between salinity, temperature, and fish energy metabolism, however, highlighting the need for further investigation in long term this area.

As a key modulator responding to cold effects, PGC-1α is claimed to regulate mitochondrial biogenesis [84] and interact with other nuclear transcriptional factors in mammals [47] and fish [85]. In mice, PGC-1α is extremely cold-sensitive due to reduced mitochondrial programs for fatty acid oxidation and electron transport in brown fat tissues caused by improper thermogenesis [86]. In the livers of the American bullfrog (*Lithobates catesbeianus*), *pgc-1α* was not activated in response to cold exposure, indicating that the ectothermic environment had no influence on their energy metabolism [87]. The wild-type fugu *(Takifugu obscurus*) boosted the energy metabolism rate in the brain and liver by oxidative phosphorylation and mitochondrial β-oxidation when the temperature was reduced to 16 °C. Moreover, glycogenolysis, lipid metabolism, and *pgc-1α* expression increased dramatically in response to cold-stress [88]. This study revealed that the relative expression of *pgc-1α* in medaka gills increased significantly in 2 h in both the FW and SW groups when exposed to 18 °C. Subsequently, *pgc-1α* expression was four times higher in the FW group and decreased to the baseline at 28 °C at 168 h in the SW group after low-temperature transfer. The gills of Indian medaka may require upregulation of *pgc-1α* to regulate mitochondrial biogenesis and increase mitochondrial density in response to cold temperatures. In addition to PGC-1α, PGC-1 coactivator family members have been identified as essential mediators of mitochondrial biogenesis [89]. PGC-1 related coactivator (PRC) is associated with mitochondrial content, which is specialized for thermogenesis in mitochondria-rich tissues [90]. PRC is involved in the expression of the respiratory chain, and its expression levels in brown fat are slightly enhanced after cold exposure for adaptive thermogenesis in mammals [91]. Following *pgc-1α* mRNA expression, the *prc* mRNA expression in the gills of FW- and SW-acclimated medaka of the cold-stress group might mitigate the effects of hypothermal stress by optimizing metabolic status as evaluated via energy reserves.

Under increasing cold-stress, stickleback (*Gasterosteus aculeatus*) muscle and liver showed increased aerobic metabolic capacity in mitochondria, including oxidative phosphorylation, fatty acid oxidation, citrate synthase (CS) activity, and cytochrome c oxidase (COX) activity [34]. Involved in oxidative phosphorylation, Nrfs trans-activate genes by modulating mtDNA transcription. As evaluated by COX activity regulated by *Nrf2* and *pgc-1α* transcriptional regulators in white muscle, it was found that several families of teleosts (*Centrarchidae*, *Umbridae*, *Esocidae*, *Gasterosteidae*, and *Cyprinidae*) increased aerobic metabolism with seasonal changes in cold winter [44]. The activation of Nrf2 (transcriptional redox sensing regulation)-dependent antioxidant heme oxygenase-1 (ho-1) expression was found to be low in the gills of Chinese Perch (*Siniperca chuatsi*) compared to that in the kidney, muscle, heart, liver, and intestine [92]. Meanwhile, in the gills of the common carp (*Cyprinus carpio*), the expression of *Nrf2* isoforms was higher when exposed to chronic Cd^2+^ stress to resist oxidative stress [93]. After exposure to 18 °C, branchial *Nrf2* gene expression responded acutely (within 24 h) in FW Indian medaka, whereas it was downregulated and then surged at 12 h in the SW group, achieving a positive correlation with *pgc-1α* and *prc*. By increasing ROS generation due to oxidative stress caused by low-temperature fluctuations, *Nrf2* gene expression in the gills of medaka can be activated, which in turn regulates mitochondrial energy metabolism.

Mitochondria are under the dual genetic control of both mitochondrial and nuclear genomes [94]. In ectothermal fish, stress-mediated responses of mtDNA are crucial for TFAM transcription. Nrf2 binding sites have been identified as upstream regulators, and PGC-1α is a common regulator of TFAM [84]. The expression of *tfam* in the gills of FW- and SW-acclimated Indian medaka did not exhibit an acute response, but elevated significantly at 168 h, after exposure to 18 °C. Knockdown of *tfam* in the zebrafish embryo resulted in a 50 ± 22% decrease in ATP production and 42 ± 17% decrease in the mtDNA copy number, leading to abnormal development of the brain, eye, muscle, and heart [58], resulting in severe depletion of mtDNA and death of embryonic respiratory chain deficit in mice [95]. Although TFAM regulation is associated with energy homeostasis, its mRNA expression may not accelerate an acute response. However, using PCR-based techniques for studying the mitochondrial genome, as well as the expression pattern of mitochondrial DNA gene expression, NADH dehydrogenase (ND5) was found to be similar to that of *tfam* which could give more evidence to the co-regulatory network for the mitochondrial genome in medaka FW and SW gills. In *Drosophila*, mitochondrial TFAM-RNAi reduced mtDNA by 40%; nonetheless, transcriptome information and mRNA levels of ND2 and ND5 remained unchanged for 8 days. It was explained, though, that TFAM was not essential for *Drosophila* mtDNA transcription [96].

Although many genes are involved in mitochondrial biogenesis via energy metabolism, hierarchical clustering is an effective interactive visualization method to find genes with similar profiles and, hence, probably related activities [97]. Exposure to 18 °C resulted in the upregulation of *pgc-1α*, *prc*, and *Nrf2*, leading to closer interconnection, which ultimately upregulates the mitochondrial genome, which is crucial for energy metabolism and ATP production, demonstrating the complex interconnectedness of genes in regulating energy metabolism in the gills of Indian medaka. Despite the nearly identical expression profiles of *tfam* and *nd5* in the FW and SW groups, *tfam* showed an abrupt decrease in the acute hypothermal stress response (2 h) in the SW group. Nevertheless, these genes can still interact with the nuclear genome to regulate ATP production.

Extreme cold-tolerance varies among teleost species. The critical cold-tolerance of the Amazon Characidae (*Brycon amazonicus*) was reported to be 10 °C [98] and that of the genetically improved farmed tilapia (GIFT, *Oreochromis niloticus*) was 10–6 °C [99]. In milkfish, the critical thermal minimum was 12.5 °C and 15.5 °C in FW- and SW-acclimated groups, respectively [14]. According to the minimal cumulative mortality, the low-temperature threshold of SW-acclimated Indian medaka has been reported to be 15 °C ([62]; Chen et al., unpublished). The extreme temperature at the lower ends of the European seabass (*Dicentrarchus labrax*) was 8 °C, their plasma osmolality levels were significantly impaired, and their plasma triglyceride, lactate, and cortisol contents increased significantly. In addition, plasma glucose and protein levels, as well as liver energy storage, decreased [100]. This study revealed that when Indian medaka were acclimated to their cold thermal extreme of 15 °C, the ATP content in the gills of the FW and SW groups increased slightly. At cold temperature extremes, anaerobic metabolism has been found to increase in some marine fishes [101], leading to the concept that animals have a reduction in the aerobic scope of the entire organism (not caused by reduced concentrations of ambient oxygen; [102]). Extreme winter temperatures increase mortality, and energy stores become limited, especially in smaller animals that have fewer reserves than larger individuals. Because of this climatic regime, it appears that metabolic energy savings are required rather than energy metabolism [103]. Specific circulatory arrangements, such as counter-current heat exchangers through the gills, help prevent the mesopelagic fish *Lampris guttatus* from losing metabolically generated heat [104]. Furthermore, exploring the mitochondrial morphological and ultrastructural features can significantly contribute to our understanding of the role of mitochondria in teleost gill metabolism, especially focusing on gill MRC mitochondria. However metabolic patterns and molecular stress responses were investigated, and the metabolic profile of fuel oxidation, when the gilthead sea bream was exposed to a critically low temperature limit (14 °C), was linked to molecular levels. The low-temperature limit stimulates anaerobic metabolism and elevates the activity of 3-hydroxyacyl CoA dehydrogenase, a process that contributes to ATP production [105]. The thermal tolerance of fish was primarily constrained by oxygen availability, as cardiovascular performance is restricted at extreme temperatures [106,107]. This limitation in fish thermal tolerance can be attributed to the initial constraint of oxygen supply capacity and the subsequent transition to anaerobic metabolism. Interestingly, hypoxia has been observed to enhance the fish’s tolerance to heat stress, from acute to chronic levels, by improving cardiovascular performance.

In the cold-tolerance experiments of Indian medaka, the expression of genes associated with mitochondrial biogenesis, such as *pgc-1α* and *prc*, increased acutely (2 h) and significantly, and subsequently reverted to and aligned with the control group. However, the FW and SW gene expression profiles did not show any significant differences over time. Phylogenetic analysis of *pgc-1α*—*prc* in the SW and FW groups in a clade further indicated the close interaction with lower divergence in the gills of Indian medaka exposed to 15 °C. PGC-1 was found to induce uncoupling and oxidative genes in mice, targeting mitochondrial biogenesis and the expression of genes involved in oxidative metabolism to generate heat (adaptive thermogenesis; [108]. Genes related to energy metabolism in brown adipose tissue (BAT) and white adipose tissue (WAT) were entirely reduced during adaptive thermogenesis in wild-type mice exposed to 4 °C for 5 h. Interestingly, the oxidative capacity and function of mitochondria in BAT and skeletal muscle are important for providing energy for thermogenesis, and there is a significant decrease in the expression levels of mitochondrial genes involved in mitochondrial biogenesis and oxidative function, such as *Nrf1*, *tfam*, and ATP synthase in the endotherms [109]. However, adaptive thermogenesis is not the principal function of ectotherms [110]. In the extreme cold-tolerance experiments of medaka, the *Nrf2* gene expression profile did not change over time.

In the cold-tolerant group of this study, downregulation of *tfam* and *nd5* was found in both FW and SW medaka, although only *tfam* expression in the FW group reached a significant level acutely between 2 h and 24 h. *tfam* interacted closely with *nd5* in the same cluster to regulate the mitochondrial genome in both FW and SW medaka. The results clearly showed that there was a different expression pattern, indicating an interaction among *nd5, tfam*, *pgc-1α*, *prc* and *Nrf1* in the cold-tolerance groups. Cluster analysis did not provide a strong connection between the nuclear and mitochondrial genomes. Extreme cold causes thermogenesis, which could upregulate metabolism for heat production rather than ATP production in ectotherms [111]. The gene expression of *pgc-1α* is a thermogenic biomarker [112], and Indian medaka might have the function of thermogenesis by upregulating *pgc-1α* and *prc* under acute cold-tolerance treatments. It is obvious, however, that mitochondrial biogenesis or mitochondrial energy metabolism did not regulate thermogenesis since the mitochondrial genome (ND5) decreased with acute cold-tolerance in the FW and SW groups.

## 4. Material and Methods

### 4.1. Experimental Fish and Environments

Indian medaka were purchased from a local agent in Taichung, Taiwan, acclimated for two weeks in brackish water (BW; 15‰), and then transferred to fresh water (FW) and seawater (SW; 35‰) at room temperature (28 ± 1 °C). A refractometer (ATC-S/Mill-E; Atago, Kyoto, Japan) was used to assess the salinity of the aerated FW after the introduction of artificial sea salt (Sea-Salt Aquarium Technology Co., Qingdao, China). The average length of an experimental fish was reared 10 weeks old, 2.5 ± 0.5 cm, sex not yet differentiated. The medaka were fed twice a day, once in the morning and once in the evening, until the day before sampling. The photoperiod was 12 h:12 h light:dark.

Accordingly, two hypothermal treatments, cold-stress (18 °C) and cold-tolerance (thermal extreme/extreme cold; 15 °C), were used in this study. In the cooling tanks, the water temperatures of the FW- and SW-acclimated Indian medaka gradually decreased from 28 °C to 18 °C and 15 °C, respectively. The water temperature was kept constant at 28 °C for the control group. The gills of six experimental fish from the control, cold-stress, and cold-tolerance groups were then sampled at various time intervals of 0, 2, 12, 24, 48, and 168 h for the following studies.

### 4.2. Determination of Cellular ATP Content in Fish Gills

Gill samples (10 mg) were homogenized in 100 μL ATP assay buffer and diluted to a concentration of 1/3 with 1× PBS before using the colorimetric test kit according to the manufacturer’s instructions (K354-100, BioVision, Milpitas, CA, USA). An ELISA microplate reader was used to obtain experimental data (VersaMax, Molecular Devices, Sunnyvale, CA, USA). For every fixed interval, readings at OD 570 nm were obtained and the time range with a steady ATP reading was studied. The ATP content was calculated using the following equation:Sample ATP concentration = B/V × D = nmol/μL
(B, the ATP amount in the reaction well from the standard curve (nmol); V, the sample volume added into sample wells (μL); D, the dilution factor).
$$\mathrm{ATP}\;\mathrm{amount}\;\mathrm{in}\;\mathrm{the}\;\mathrm{sample}\;\mathrm{well}\;(B)=\frac{OD\;sample}{\left(OD\;sample+ATP\;std\right)-(OD\;sample)}\;*\;\mathrm{ATP}\;\mathrm{Standards}\;(\mathrm{pmol})$$

### 4.3. Preparation of Total RNA Samples from Fish Gills

Total RNA was extracted using TriPure Isolation Reagents (#11667165001; Roche, Mannheim, Germany) and homogenized using a POLYTRON (PT1200E; Kinematica, Lucerne, Switzerland) at maximal speed for 10 s/strokes. After anesthetization, the experimental fish were euthanized, their gill covers were removed, and the first gill of each side was sampled and put into a 1.5 mL microcentrifuge tube containing 200 µL TriPure. The tubes containing the gill samples were treated with liquid nitrogen and then stored at −80 °C for subsequent experiments. RNA extraction was performed according to [113]. RNA quality was checked using Nanodrop 2000 (Thermo, Wilmington, DE, USA). The values of A260/A280 and A260/A230 were confirmed to be 1.8–2.0 and ~2.0, respectively. Then, a nucleic acid stain (SafeView™ Classic; ABM, San Jose, CA, USA) in 1% Agarose gel (SeaKem^®^ LE Agarose; Lonza, Basel, Switzerland) and sample stain (6× Loading Dye; BV110; BioVan, Taichung, Taiwan) were used to perform gel electrophoresis.

### 4.4. The cDNA (Complementary DNA) Preparation and qPCR Analysis

First, general cDNA was prepared using reverse transcription polymerase chain reaction following the manufacturer’s instructions (iScriptTM Reverse Transcriptase; Bio-Rad, Hercules, CA, USA). One microgram of extracted RNA was used for cDNA synthesis using a thermal cycler (Applied Biosystems^®^ Veriti 96-Well Thermal Cycler; Thermo, San Francisco, CA, USA). Quantitative PCR analysis was performed using a MiniOpticon real-time PCR system (Bio-Rad, CA, USA). The sequences of the primer pairs used for the qPCR analyses are listed in Table 1. At the same time, 10 μL of 2× FastStart Universal SYBR Green Master (04913850001; Roche, Mannheim, Germany), 2 μL cDNA, and 1 μL primer pair (250 nM) were combined to a total volume of 20 μL with sterile water.

Real-time PCR was performed using the following steps: (1) 95 °C for 10 min, (2) 95 °C for 15 s, and (3) 60 °C for 45 s, with a total of 40 cycles. Relative quantification of gene expression was performed by comparing the expression between the target genes and the housekeeping gene (*β-actin*). The expression level of the target gene was calculated using the comparative Ct (threshold cycle number) method with the formula 2^ ^− [(Ct target gene, n—Ct β-actin, n)—(Ct target gene, c—Ct β-actin, c)]^ [114], where n is the unknown sample and c is the control group sample.

### 4.5. Statistical Analysis

Statistical analyses were performed using the Minitab 16 program (Minitab, State College, PA, USA), and the normality test was performed using the Anderson-Darling test. Mann-Whitney pairwise comparisons were used as non-parametric correlation tests. The mean ± standard error of the mean (SEM) was used to express the values, and the significance level was set at *p* < 0.05.

### 4.6. Hierarchical Clustering Analysis

We sought to determine if groupings of genes have a strong correlation to proximal gene expression/functions, as in higher vertebrates. Related genes should be found in the same cluster, and the hierarchical clustering algorithm should locate similar groupings [97]. This will result in understanding which unknown genes cluster with known genes, which may aid in our understanding of the functions of unknown genes. The use of precise color mosaics aids in the discovery of these linkages [115].

Using the qPCR gene expression of distinct experimental groups, hierarchical clustering of gene expression was carried out [115]. The data were logarithmically transformed during the hierarchical cluster analysis in the Gene Cluster 3.0 software. The data were then transformed using the Pearson approach. After grouping the relevance analysis, heat maps were created using Java Treeview software v3.0 and the complete linkage approach.

## 5. Conclusions

This study improved our understanding of the responses of mitochondrial energy metabolism in gills to different levels of hypothermal challenges by exposing Indian medaka to cold-stress and cold-tolerance (thermal extreme) environments in FW or SW over a period of time (168 h). When the low temperatures varied through cold-stress until the cold-tolerance level was reached, *pgc-1α* regulated mitochondrial biogenesis to energy metabolism in various ways by regulating *prc*. Under cold-stress, *pgc-1α* has the ability to activate nuclear transition via mitochondrial energy metabolism for ATP production. When fish were exposed to a cold-tolerant environment, the *Nrf2* function in the gills was neutralized, and they were unable to regulate mitochondrial energy homeostasis followed by downregulation of mitochondrial-related genes and reduced ATP production.

## Figures and Tables

**Figure 1 ijms-24-16187-f001:**
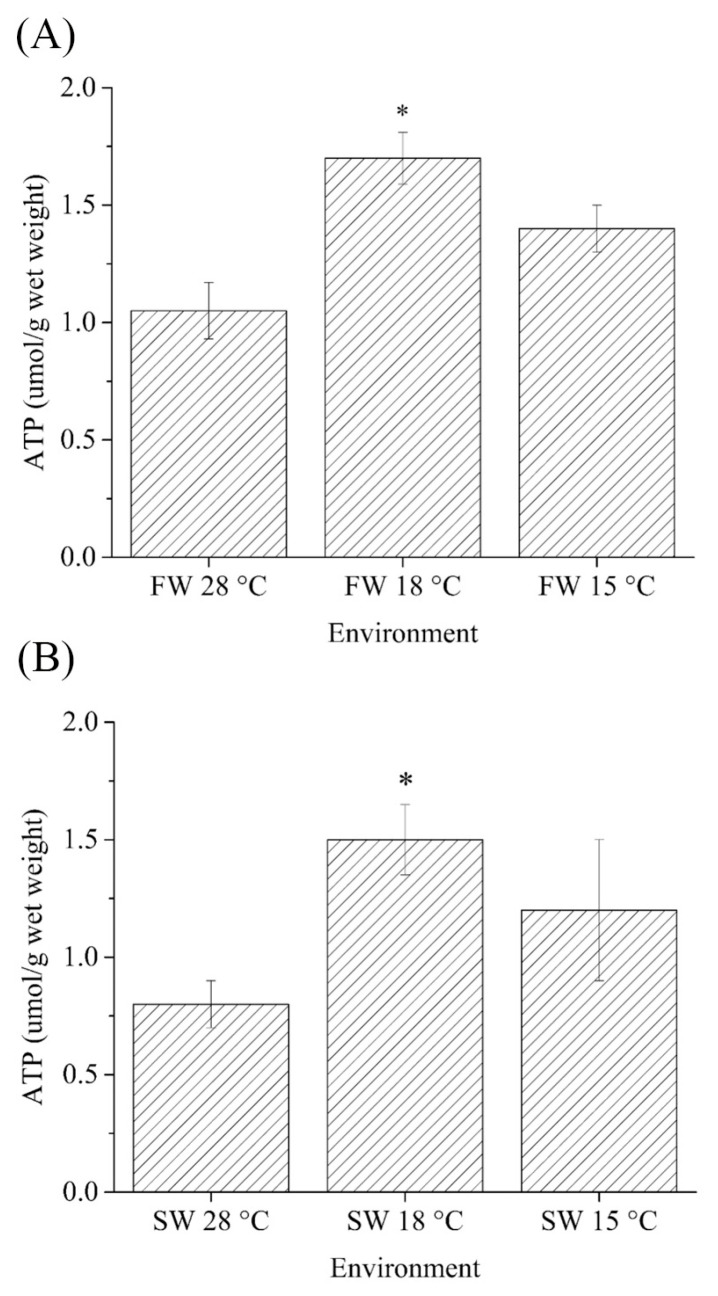
Comparisons of ATP contents (μmoL/g) in gills of (**A**) fresh water (FW)- and (**B**) seawater (SW)-acclimated Indian medaka at 28, 18 and 15 °C for one week. The asterisk indicate a significant difference (*n* = 6, Mann-Whitney test, *p* < 0.05).

**Figure 2 ijms-24-16187-f002:**
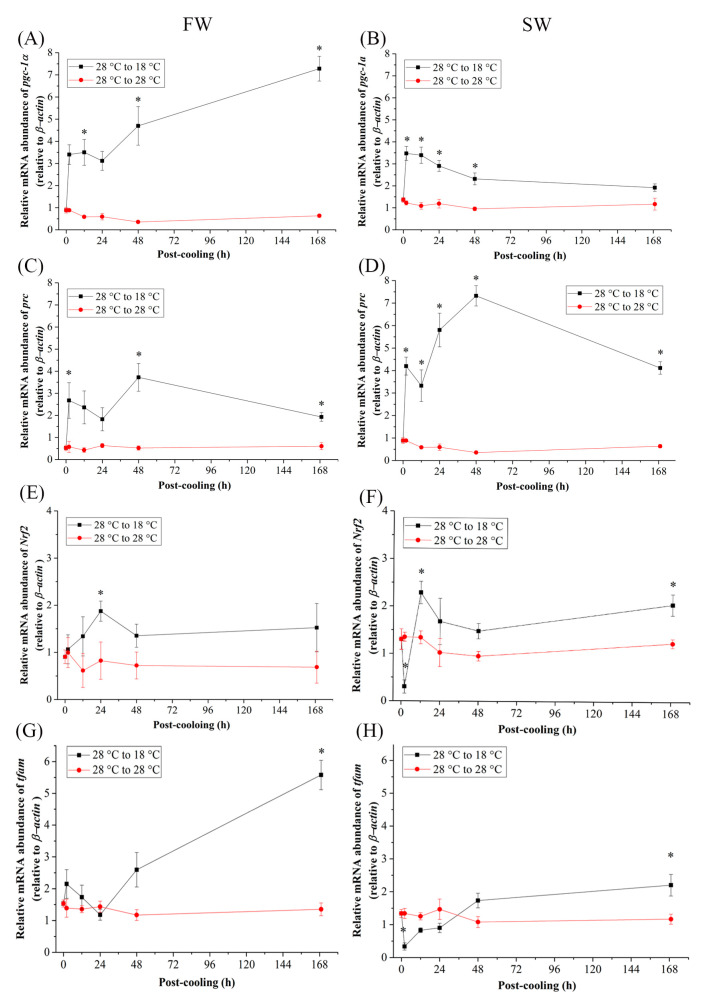
Relative mRNA expression of (**A**,**B**) *pgc-1α*, (**C**,**D**) *prc*, (**E**,**F**) *Nrf2*, and (**G**,**H**) *tfam* in gills of fresh water (FW)- and seawater (SW)-acclimated Indian medaka transferred from 28 °C to 28 °C, and cooled down at 18 °C for one week. The expression has been normalized by the expression of the *β-actin*. The asterisks (*) indicated a significant difference to each time point in the control group. (*n* = 6; Mann-Whitney test, *p* < 0.05).

**Figure 3 ijms-24-16187-f003:**
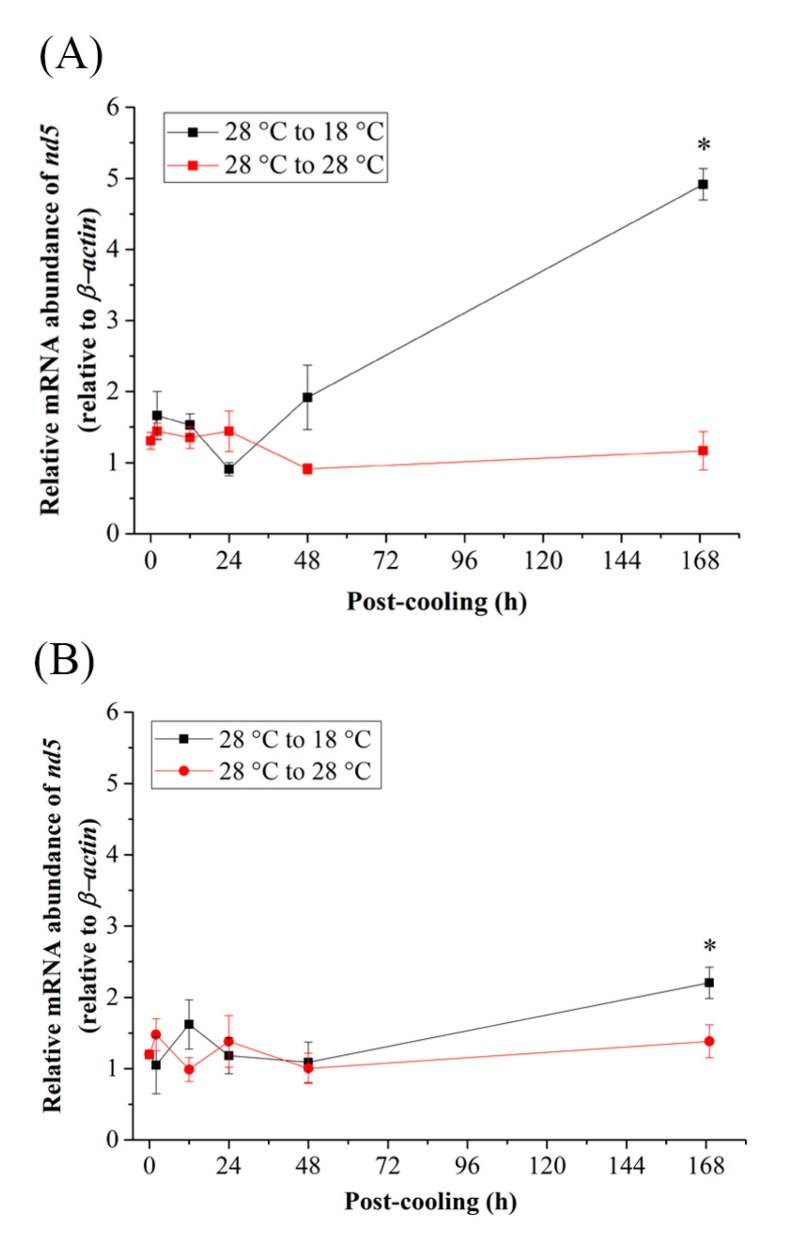
Relative mRNA expression of *nd5* in gills of (**A**) fresh water (FW)- and (**B**) seawater (SW)-acclimated Indian medaka transferred from 28 °C to 28 °C, and cooled down at 18 °C for one week. The expression has been normalized by the expression of *β-actin*. The asterisks (*) indicated a significant difference to each time point in the control group. (*n* = 6; Mann-Whitney test, *p* < 0.05).

**Table 1 ijms-24-16187-t001:** Primer sequences used for the cDNA cloning (PCR) and expression detection (qPCR) of medaka gills.

Gene		Primer Sequence (5′ to 3′)	Amplicon Size (bp)	Reference Number	Primer Concentration
*pgc-1α*	F:	ACCTACCGCTATACTTGTGAC	130	XM_024287565.2	250 nM.
	R:	AAGTCTGTGTAATGGGATTTGC			
*prc*	F:	CAGTAGAGACTAAAGACGAGGAG	138	XM_024297615.2	250 nM.
	R:	CTGCCCTTCTGATTAGACAGC			
*Nrf2*	F:	GCAAGTGGATGAAACGATGGA	102	XM_036212722.1	250 nM.
	R:	GGGAGTATTCTGGATTAACTGGT			
*Tfam*	F:	GGAGGAGTCTGTTCAGCAACCAG	124	XM_024298155.2	250 nM
	R:	GAGGTCTTCTCGTCCGATCTCCA			
*nd5*	F:	CCACCCACGATTTAATTCACTC	110	NC_012976.1	250 nM
	R:	TAATAGTCCCGCTAAGATACTTCC			
*β-actin*	F:	CCATTGAGCACGGTATTGTCA	102	XM_024296129.1	250 nM
	R:	GCAACACGCAGCTCGTTGTA			

## Data Availability

Data are contained within the article.

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
