# Peer review of "Regulation of PGC-1α of the Mitochondrial Energy Metabolism Pathway in the Gills of Indian Medaka (Oryzias dancena) under Hypothermal Stress"

_ijms, 2023, doi:10.3390/ijms242216187_

Round 1

Reviewer 1 Report

Comments and Suggestions for Authors

The manuscript presents an interesting issue evaluating the water temperature related induced stress in a marine fish that serves as model species. The methodology is correctly described and the text is written in a good manner. My main criticism concerns the genes examined, which although well explained, are not sufficient for such a study. Particularly, since temperature related stress is investigated, such genes should have been examined, perhaps in combination with oxidative stress related genes, on account of the correlation of dissolved oxygen with temperature. At least comparison with HSPs or catalase. Unless a reasonable explanation is provide for this fact, I cannot recommend publication.

Specific comments

The Introduction is generally well written, including detailed information for various genes affected by environmental factors. Nevertheless, some more details for the medaka are essential. How is it influenced by temperature or dissolved oxygen? Some more info has to be added.

Twenty independent regions…: I believe this statement is not correct. The number of mitochondrial genes is 13 whereas there are additionally more than 20 tRNAs, so in total more than 30 genes. Also, the first half paragraph is irrelevant and should be deleted.

The discussion needs restructuring. The first two paragraphs have to be combined in order to firstly present and commentary the findings and then compare with literature data. 

The third paragraph discusses how salinity could have affected the energy demand and metabolism. I believe the it is written is seems out of the scope. Maybe if reorganised the inferences could be better justified. As it is it does not come out that cold stress increases energy metabolism 

Author Response

The manuscript presents an interesting issue evaluating the water temperature related induced stress in a marine fish that serves as model species. The methodology is correctly described and the text is written in a good manner. My main criticism concerns the genes examined, which although well explained, are not sufficient for such a study. Particularly, since temperature related stress is investigated, such genes should have been examined, perhaps in combination with oxidative stress related genes, on account of the correlation of dissolved oxygen with temperature. At least comparison with HSPs or catalase. Unless a reasonable explanation is provide for this fact, I cannot recommend publication.

Answer:

Dear Reviewer,

Thank you for your valuable feedback on our study investigating for our manuscript. We appreciate your thoughtful comments and would like to address your concerns regarding the selection of genes for our study. We understand your point regarding the importance of considering genes related to oxidative stress and other factors, such as HSPs (Heat Shock Proteins) or catalase, especially in the context of temperature-related stress, given the correlation of dissolved oxygen with temperature. We acknowledge that these genes play crucial roles in stress responses and metabolic regulation.

The genes we selected for our study were chosen based on their well-established roles in mitochondrial energy metabolism in higher vertebrates and their relevance to the specific research question we aimed to address in fishe under different cold stresses. However, we acknowledge that including additional genes related to oxidative stress and other stress response pathways would provide a more comprehensive view of the adaptive mechanisms in euryhaline fish under changing environmental conditions. In our study, we focused on elucidating the role of pgc-1α and Nrf2 in nuclear regulation to mitochondrial energy metabolism, and their interactions with downstream genes, as these factors are known to be key regulators of mitochondrial function and ATP production. Nonetheless, we recognize the importance of considering a broader range of genes in future research to better understand the multifaceted responses of fish to environmental stressors.

To address your concern and enhance the comprehensiveness of our study, we plan to conduct further research, which will include an analysis of genes related to oxidative stress and other stress response pathways. We believe that this approach will provide a more holistic understanding of the adaptive strategies employed by euryhaline fish in response to changing environmental conditions. Furthermore, we will introduce a discussion on dissolved oxygen with temperature and the regulation of HSPs, which will be important for guiding future experiments.

We hope that these explanations address your concerns and demonstrate our commitment to enhancing the quality and completeness of our research more. We appreciate your constructive feedback and will work diligently to incorporate your suggestions into our future work.

Specific comments

  1. The Introduction is generally well written, including detailed information for various genes affected by environmental factors. Nevertheless, some more details for the medaka are essential. How is it influenced by temperature or dissolved oxygen? Some more info has to be added.

Answer: Thank you for your question. The introduction section has been updated to include information on how temperature and dissolved oxygen can affect the Indian medaka's increased anaerobic metabolism. However, more detailed information related to euryhaline fish anaerobic glycolysis and hypoxic conditions has been included in the discussion section

Ref:

  • Li, A. J.; Leung, P. T.; Bao, V. W.; Lui, G. C.; Leung, K. M., Temperature-dependent physiological and biochemical responses of the marine medaka Oryzias melastigma with consideration of both low and high thermal extremes. Journal of Thermal Biology 2015, 54, 98-105.
  • Jackson, S. E., Hsp90: structure and function. Molecular chaperones 2013, 155-240.
  1. Twenty independent regions…: I believe this statement is not correct. The number of mitochondrial genes is 13 whereas there are additionally more than 20 tRNAs, so in total more than 30 genes. Also, the first half paragraph is irrelevant and should be deleted.

Answer: Thank you for your comment. This sentence and others have been rewritten. The first half paragraph is deleted.

  1. The discussion needs restructuring. The first two paragraphs have to be combined in order to firstly present and commentary the findings and then compare with literature data.

Answer: Thank you for your comment. According to the review comment first two paragraphs have combined in order to first present and commentary elucidated

  1. The third paragraph discusses how salinity could have affected the energy demand and metabolism. I believe the it is written is seems out of the scope. Maybe if reorganised the inferences could be better justified. As it is it does not come out that cold stress increases energy metabolism

Answer: Thank you for your comment. The third paragraph of the discussion section has been revised to align with the scope of the study and explain how various euryhaline fish species may differ in their energy demands and metabolism in cold acclimation.

Reviewer 2 Report

Comments and Suggestions for Authors

The aim of this manuscript is to apply two hypothermal treatments to reveal the mechanisms of energy metabolism via pgc-1α regulation in the gills of Indian medaka (Oryzias dancena): cold-stress (18°C) and cold-tolerance (extreme cold;15°C).

This manuscript shows rich content, providing a deep insight for some works: the study is within the journal’s scope, and I found it to be well-written, providing sufficient information. Even if the manuscript provides an organic overview, with a densely organized structure and based on well-synthetized evidence, there are some suggestions necessary to make the article complete and fully readable. For these reasons, the manuscript requires major changes.

Please find below an enumerated list of comments on my review of the manuscript:

Please, the authors should provide a list of the abbreviations, mentioned in this manuscript.

INTRODUCTION:

In this introductive section, the manuscript may benefit from reporting that recent studies suggested that water temperature may also be associated to the onset and progression of specific disease, in fishes, by influencing pathogen transfer and disease development (see, for reference: https://doi.org/10.1016/j.aquaculture.2022.738577).

Furthermore, ultrastructural studies also reported the presence of mitochondrial rich cells (MRCs), as a fundamental component of the surface layer of the gill epithelium (see, for reference: https://doi.org/10.1002/jmor.20757).

DISCUSSION:

The authors should mention, in this discussive section, the future perspectives of the study. Among them, a significative contribute to the exploration of mitochondrial role in teleost metabolism might be provided by ultrastructural evidence (SEM and TEM).

The main topic is interesting, and certainly of great clinical impact. As regards the originality and strengths of this manuscript, this is a significant contribute to the ongoing research on this topic, as it extends the research field on the mechanisms of energy metabolism via pgc-1α regulation in the gills of Indian medaka (Oryzias dancena): cold-stress (18°C) and cold-tolerance (extreme cold;15°C). Overall, the contents are rich, and the authors also give their deep insight for some works.

As regards the section of methods, there is a specific and detailed explanation for the methods used in this study: this is particularly significant, since the manuscript relies on a multitude of methodological and statistical analysis, to derive its conclusions. The methodology applied is overall correct, the results are reliable and adequately discussed.

The conclusion of this manuscript is perfectly in line with the main purpose of the paper: the authors have designed and conducted the study properly. As regards the conclusions, they are well written and present an adequate balance between the description of previous findings and the results presented by the authors.

Finally, this manuscript also shows a basic structure, properly divided and looks like very informative on this topic. Furthermore, figures and tables are complete, organized in an organic manner and easy to read.

In conclusion, this manuscript is densely presented and well organized, based on well-synthetized evidence. The authors were lucid in their style of writing, making it easy to read and understand the message, portrayed in the manuscript. Besides, the methodology design was appropriately implemented within the study. However, many of the topics are very concisely covered. This manuscript provided a comprehensive analysis of current knowledge in this field. Moreover, this research has futuristic importance and could be potential for future research. However, major concerns of this manuscript are with the introductive and discussive sections: for these reasons, I have major comments for these sections, for improvement before acceptance for publication. The article is accurate and provides relevant information on the topic and I have some major points to make, that may help to improve the quality of the current manuscript and maximize its scientific impact. I would accept this manuscript if the comments are addressed properly.

Comments on the Quality of English Language

Minor editing of English Language are necessary.

Author Response

The aim of this manuscript is to apply two hypothermal treatments to reveal the mechanisms of energy metabolism via pgc-1α regulation in the gills of Indian medaka (Oryzias dancena): cold-stress (18°C) and cold-tolerance (extreme cold;15°C).

This manuscript shows rich content, providing a deep insight for some works: the study is within the journal’s scope, and I found it to be well-written, providing sufficient information. Even if the manuscript provides an organic overview, with a densely organized structure and based on well-synthetized evidence, there are some suggestions necessary to make the article complete and fully readable. For these reasons, the manuscript requires major changes.

Please find below an enumerated list of comments on my review of the manuscript:

Please, the authors should provide a list of the abbreviations, mentioned in this manuscript.

Answer: We would like to thank the reviewer for his thorough revision by providing comments and suggestions for our manuscript. Abbreviations are updated on the manuscript's first page.

INTRODUCTION:

In this introductive section, the manuscript may benefit from reporting that recent studies suggested that water temperature may also be associated to the onset and progression of specific disease, in fishes, by influencing pathogen transfer and disease development (see, for reference: https://doi.org/10.1016/j.aquaculture.2022.738577)

Answer: Thank you for your comment. According to your suggestion, we have updated the references in the first paragraph in the introduction.

Furthermore, ultrastructural studies also reported the presence of mitochondrial rich cells (MRCs), as a fundamental component of the surface layer of the gill epithelium (see, for reference: https://doi.org/10.1002/jmor.20757).

Answer: Thank you for your comment. We have followed the reference in the introduction. Please notice the second paragraph in the introduction.

 DISCUSSION:

The authors should mention, in this discussive section, the future perspectives of the study. Among them, a significative contribute to the exploration of mitochondrial role in teleost metabolism might be provided by ultrastructural evidence (SEM and TEM).

Answer: Thank you for your comment. We have updated the future perspectives of the study in the seventh paragraph of the discussion.

The main topic is interesting, and certainly of great clinical impact. As regards the originality and strengths of this manuscript, this is a significant contribute to the ongoing research on this topic, as it extends the research field on the mechanisms of energy metabolism via pgc-1α regulation in the gills of Indian medaka (Oryzias dancena): cold-stress (18°C) and cold-tolerance (extreme cold;15°C). Overall, the contents are rich, and the authors also give their deep insight for some works.

As regards the section of methods, there is a specific and detailed explanation for the methods used in this study: this is particularly significant, since the manuscript relies on a multitude of methodological and statistical analysis, to derive its conclusions. The methodology applied is overall correct, the results are reliable and adequately discussed.

The conclusion of this manuscript is perfectly in line with the main purpose of the paper: the authors have designed and conducted the study properly. As regards the conclusions, they are well written and present an adequate balance between the description of previous findings and the results presented by the authors.

Finally, this manuscript also shows a basic structure, properly divided and looks like very informative on this topic. Furthermore, figures and tables are complete, organized in an organic manner and easy to read.

In conclusion, this manuscript is densely presented and well organized, based on well-synthetized evidence. The authors were lucid in their style of writing, making it easy to read and understand the message, portrayed in the manuscript. Besides, the methodology design was appropriately implemented within the study. However, many of the topics are very concisely covered. This manuscript provided a comprehensive analysis of current knowledge in this field. Moreover, this research has futuristic importance and could be potential for future research. However, major concerns of this manuscript are with the introductive and discussive sections: for these reasons, I have major comments for these sections, for improvement before acceptance for publication. The article is accurate and provides relevant information on the topic and I have some major points to make, that may help to improve the quality of the current manuscript and maximize its scientific impact. I would accept this manuscript if the comments are addressed properly.

Answer: We would like to thank you for providing comments and suggestions for our manuscript. We have modified the writing of the manuscript according to the reviewer’s recommendation.

Reviewer 3 Report

Comments and Suggestions for Authors

This is a study for which the general interest is not put forward, and we don't understand (apart from the contribution of fundamental knowledge) the interest of studying this genetic regulation in this fish and how this is an important scientific contribution.

The article is very descriptive in its presentation of data, and offers little in the way of progress on a new function, particularly for the species studied.

Comments on the Quality of English Language

There are several small English errors in the text (typography?).

A careful re-reading should enable us to correct them.

Author Response

Comments and Suggestions for Authors

This is a study for which the general interest is not put forward, and we don't understand (apart from the contribution of fundamental knowledge) the interest of studying this genetic regulation in this fish and how this is an important scientific contribution.

Answer: We would like to thank the reviewer for his thorough revision by providing comments and suggestions for our manuscript. We have modified the writing of the manuscript according to the reviewer’s recommendation to emphasizing the significant scientific contribution of this study to the field of fish research in the last paragraph of the abstract and introduction.

The article is very descriptive in its presentation of data, and offers little in the way of progress on a new function, particularly for the species studied.

Answer: Thank you for your comment. We attempted to describe the studies which have been conducted on the effect of salinity and temperature on energy metabolism using euryhaline model species. However, in the abstract, we have updated how this study provides insights into the complex ways in which fish adjust to cold stress, contributing to our knowledge.

Comments on the Quality of English Language

There are several small English errors in the text (typography?). A careful re-reading should enable us to correct them.

Answer: Thank you for your suggestion. We have checked and corrected several English errors for clarity.

Reviewer 4 Report

Comments and Suggestions for Authors

comments are attached 

Comments on the Quality of English Language

A through revision is required for language in terms of grammar and phrases

Author Response

Regulation of PGC-1α of the mitochondrial energy metabolism pathway in the gills of Indian medaka (Oryzias dancena) under hypothermal stress

Abstract

  • The authors could provide more information about the fish that were used in the study, such as their age, size, and sex.

Answer:  Thank you for your valuable feedback on our study investigation for the manuscript. We appreciate your thoughtful comments.  We have updated the information in the abstract and material and methods.

  • The authors could also provide more information about the experimental conditions, such as the photoperiod and the feeding regime.

Answer:  Thank you for your comment. We have updated the information.

  • The authors could discuss the implications of their findings for other ectothermic fish species and for the potential impacts of climate change on ectothermic fish populations.

Answer:  Thank you for your comment. We have updated the information about different hypothermal stress-induced profiles and durations in different fish species.

Introduction

  • The authors could provide more information about the specific hypothermal stress treatments that were used in the studies that they reviewed. For example, what were the specific temperatures and durations of exposure?

Answer:  Thank you for your comment. We have updated the information about low hypothermal stress and durations of exposure in different species in the first paragraph of the introduction.

  • The authors could also discuss the limitations of the existing literature and suggest areas for future research. For example, there is a need for more studies on the long-term effects of hypothermal stress on mitochondrial energy metabolism in fish gills.

Answer: 

Thank you for your comment.

The limitation of this study is emphasized as, “more studies are required to address the dearth of information and the inherent ambiguity in understanding how mitochondrial biogenesis and energy metabolism intersect in euryhaline fish during cold exposure, emphasizing the pressing need to investigate the long-term effects of hypothermal stress on fish gill mitochondria” in introduction.

For the limitation of this study about “thermal tolerance of fish was primarily constrained by oxygen availability, as cardiovascular performance is restricted at extreme temperatures. This limitation in fish thermal tolerance can be attributed to the initial constraint of oxygen supply capacity and the subsequent transition to anaerobic metabolism. Interestingly, hypoxia has been observed to enhance the fish's tolerance to heat stress, from acute to chronic levels, by improving cardiovascular performance” updated in discussion.

References:

  • 1) Mendonça, P. C.; Gamperl, A. K., The effects of acute changes in temperature and oxygen availability on cardiac performance in winter flounder (Pseudopleuronectes americanus). Comparative Biochemistry and Physiology Part A: Molecular & Integrative Physiology 2010, 155, (2), 245-252.
  • 2) Pörtner, H.-O., Oxygen-and capacity-limitation of thermal tolerance: a matrix for integrating climate-related stressor effects in marine ecosystems. Journal of Experimental Biology 2010, 213, (6), 881-893.

  • The authors could proofread the review carefully to catch any minor errors in grammar or spelling.

Answer: Thank you for your suggestion. Our manuscript was English edited by a native speaker from a company providing English editing service. We have attached the pdf of the editing certificate for your reference. Please find the attachment. Further we have checked and corrected several English minor errors for clarity.

  • In the first paragraph, the authors state that "The gill is one of the primary organs for osmoregulation in fish and is sensitive to physical and chemical changes in aquatic environments." It would be helpful to provide some specific examples of these physical and chemical changes.

Answer: Thank you for your suggestion. We have updated the examples for the physical and chemical changes in aquatic environments.

  • In the second paragraph, the authors discuss the role of PGC-1α in the control of energy metabolism. It would be helpful to provide a more detailed explanation of how PGC-1α regulates mitochondrial biogenesis and energy production.

Answer: Thank you for your suggestion. We have updated the role of PGC-1α in the control of energy metabolism, “PGC-1α serves as a central regulator of mitochondrial biogenesis and energy production by partnering with deacetylases and phosphorylases to bind nuclear receptors. This collaboration results in the upregulation of various transcription factors and subsequent expression of proteins encoded by both nuclear and mitochondrial genomes, particularly those associated with the oxidative phosphorylation complexes (OXPHOS) in the mitochondria. This orchestration, led by PGC-1α, promotes the efficient production of ATP, a crucial factor in energy generation, and underlines its role as a master regulator of this process. Furthermore, PGC-1α's activity is closely intertwined with AMPK, a sensor of cellular energy status, which further emphasizes its critical role in regulating mitochondrial biogenesis and oxidative metabolism”. 

References:

  • Kong, S.; Cai, B.; Nie, Q., PGC-1α affects skeletal muscle and adipose tissue development by regulating mitochondrial biogenesis. Molecular Genetics and Genomics 2022, 297, (3), 621-633.
  • Fuentes, E. N.; Safian, D.; Einarsdottir, I. E.; Valdés, J. A.; Elorza, A. A.; Molina, A.; Björnsson, B. T., Nutritional status modulates plasma leptin, AMPK and TOR activation, and mitochondrial biogenesis: implications for cell metabolism and growth in skeletal muscle of the fine flounder. General and Comparative Endocrinology 2013, 186, 172-180.

  • In the third paragraph, the authors discuss the role of Nrf2 in the regulation of oxidative stress. It would be helpful to provide a more detailed explanation of how Nrf2 activates the Nrf2-TFAM pathway to improve mitochondrial dysfunction and promote mitochondrial biosynthesis.

Answer: Thank you for your suggestion. We have updated how Nrf2 activates the Nrf2-TFAM pathway and mitochondrial regulation; [48]. In non-stressed cells, Nrf2 is suppressed by Keap1 , promoting its degradation, while in stressed cells, the disruption of this interaction activates Nrf2, enabling it to enter the nucleus and function as a transcriptional activator by binding to antioxidant response elements (AREs) in the DNA, thus allowing the transcriptional activation of its target genes [49, 50]. Moreover, Nrf2 plays a role in the activation of mitochondrial biogenesis by binding to adenosine and uridine-rich elements (AREs) in the NRF1 promoter, ultimately leading to the activation of mitochondrial transcription factor A (TFAM)[51, 52].

References:

  • Hayes, J. D.; Ebisine, K.; Sharma, R. S.; Chowdhry, S.; Dinkova-Kostova, A. T.; Sutherland, C., Regulation of the CNC-bZIP transcription factor Nrf2 by Keap1 and the axis between GSK-3 and β-TrCP. Current Opinion in Toxicology 2016, 1, 92-103.
  • McMahon, M.; Itoh, K.; Yamamoto, M.; Hayes, J. D., Keap1-dependent proteasomal degradation of transcription factor Nrf2 contributes to the negative regulation of antioxidant response element-driven gene expression. Journal of Biological Chemistry 2003, 278, (24), 21592-21600.
  • Chen, Z.; Wang, H.; Hu, B.; Chen, X.; Zheng, M.; Liang, L.; Lyu, J.; Zeng, Q., Transcription factor nuclear factor erythroid 2 p45-related factor 2 (NRF2) ameliorates sepsis-associated acute kidney injury by maintaining mitochondrial homeostasis and improving the mitochondrial function. European Journal of Histochemistry: EJH 2022, 66, (3).
  • Wu, K. L.; Wu, C.-W.; Chao, Y.-M.; Hung, C.-Y.; Chan, J. Y., Impaired Nrf2 regulation of mitochondrial biogenesis in rostral ventrolateral medulla on hypertension induced by systemic inflammation. Free Radical Biology and Medicine 2016, 97, 58-74.

Results

  • The authors have done a good job of introducing the topic and explaining the rationale for the study.
  • The methods are well-described and the results are presented in a clear and concise manner.
  • The authors have discussed the implications of their findings for understanding how Indian medaka cope with cold stress and how these fish may be affected by climate change.

  • The authors could provide more information about the fish that were used in the study, such as their age, size, and sex.

Answer:  Thank you for your comment. We have updated information about the fish in material and methods.

  • The authors could also discuss the limitations of their study and suggest areas for future research. For example, it would be interesting to investigate the long-term effects of cold stress on mitochondrial energy metabolism in Indian medaka gills.

Answer: 

Thank you for your comment.

The limitation of this study is emphasized as, “more studies are required to address the dearth of information and the inherent ambiguity in understanding how mitochondrial biogenesis and energy metabolism intersect in euryhaline fish during cold exposure, emphasizing the pressing need to investigate the long-term effects of hypothermal stress on fish gill mitochondria” in introduction.

Discussion

  • The author's discussion of the implications of their findings is superficial and does not adequately address the limitations of the study.

Answer: 

Thank you for your comment.

The limitation of this study about “thermal tolerance of fish was primarily constrained by oxygen availability, as cardiovascular performance is restricted at extreme temperatures. This limitation in fish thermal tolerance can be attributed to the initial constraint of oxygen supply capacity and the subsequent transition to anaerobic metabolism. Interestingly, hypoxia has been observed to enhance the fish's tolerance to heat stress, from acute to chronic levels, by improving cardiovascular performance” updated in the discussion.

References:

  • Mendonça, P. C.; Gamperl, A. K., The effects of acute changes in temperature and oxygen availability on cardiac performance in winter flounder (Pseudopleuronectes americanus). Comparative Biochemistry and Physiology Part A: Molecular & Integrative Physiology 2010, 155, (2), 245-252.
  • Pörtner, H.-O., Oxygen-and capacity-limitation of thermal tolerance: a matrix for integrating climate-related stressor effects in marine ecosystems. Journal of Experimental Biology 2010, 213, (6), 881-893.

Overall

Overall, this is a well-written and informative review article that makes a significant contribution to our understanding of the regulation of PGC-1α of the mitochondrial energy metabolism pathway in the gills of Indian medaka under hypothermal stress. I recommend that this article be published in a peer-reviewed journal. However, still some of the above-mentioned points needs to be addressed before it can be processed further. Author needs to increase the resolution of figures and a through revision for its language is highly essential. I recommends it major revision.

Round 2

Reviewer 1 Report

Comments and Suggestions for Authors

I am generally satisfied with the corrections performed. I would recommend however to include the explanation for the selection of the genes in  the introduction. Then the manuscript can be published

Author Response

I am generally satisfied with the corrections performed. I would recommend however to include the explanation for the selection of the genes in the introduction. Then the manuscript can be published

Answer: Thank you for your valuable feedback on our study investigation of our manuscript. We appreciate your thoughtful comments and would like to address your concerns regarding the selection of explanations for the selection of genes for our study, we have explained what is the importance of each gene separately. However, we have updated how genes are selected via their regulation pathway in the introduction.

References: 

LeMoine, C. M.; Genge, C. E.; Moyes, C. D., Role of the PGC-1 family in the metabolic adaptation of goldfish to diet and temperature. Journal of Experimental Biology 2008, 211, (9), 1448-1455.

Reviewer 2 Report

Comments and Suggestions for Authors

The authors should provide recent and updated evidence on the effect of the water temperature on the onset and progression of specific disease, as suggested by recent morpho-functional data. 

Comments on the Quality of English Language

Minor editing of English language  are required.

Author Response

The authors should provide recent and updated evidence on the effect of the water temperature on the onset and progression of specific diseases, as suggested by recent morpho-functional data.

Answer: We would like to thank the reviewer for his comments and suggestions. To provide evidence on the effect of the water temperature and specific diseases related to the gills and their recent morpho-functional data are limited. However, we have updated in the introduction several temperature effects on gill pathology and also explained how low temperature could change gill plasticity lead to changes using recent data.

References:

  1. Benedicenti, O.; Pottinger, T. G.; Collins, C.; Secombes, C. J., Effects of temperature on amoebic gill disease development: Does it play a role? Journal of fish diseases 2019, 42, (9), 1241-1258.
  2. Gibbons, T. C.; McBryan, T. L.; Schulte, P. M., Interactive effects of salinity and temperature acclimation on gill morphology and gene expression in threespine stickleback. Comparative Biochemistry and Physiology Part A: Molecular & Integrative Physiology 2018, 221, 55-62.
  3. Nilsson, G. E.; Dymowska, A.; Stecyk, J. A., New insights into the plasticity of gill structure. Respiratory physiology & neurobiology 2012, 184, (3), 214-222.

Reviewer 4 Report

Comments and Suggestions for Authors

Article can be accepted .

Comments on the Quality of English Language

Minor changes are required. 

Author Response

Answer: Thank you for your suggestion. We have checked and corrected several English errors for clarity.
